# Synthesis and Luminescence Properties of Core-Shell-Shell Composites: SiO_2_@PMDA-Si-Tb@SiO_2_ and SiO_2_@PMDA-Si-Tb-phen@SiO_2_

**DOI:** 10.3390/nano9020189

**Published:** 2019-02-02

**Authors:** Lina Feng, Wenxian Li, Jinrong Bao, Yushan Zheng, Yilian Li, Yangyang Ma, Kuisuo Yang, Yan Qiao, Anping Wu

**Affiliations:** 1Inner Mongolia Key Laboratory of Chemistry and Physics of Rare Earth Materials, School of Chemistry and Chemical Engineering, Inner Mongolia University, Hohhot 010021, China; nmgfln@163.com (L.F.); jinrongbao@imu.edu.cn (J.B.); yilian005@126.com (Y.L.); ma1034537689@163.com (Y.M.); nmyangks@163.com (K.Y.); 15714848705@163.com (Y.Q.); 15548098722@163.com (A.W.); 2Inner Mongolia Autonomous Region Food Inspection Test Center, Hohhot 010010, China; nmgzys@126.com

**Keywords:** 2,5-bis((3-(triethoxysilyl)propyl)carbamoyl)terephthalic acid (PMDA-Si), core-shell composite, core-shel-shell composite, silica shell, photoluminescence, lifetime

## Abstract

Two novel core-shell composites SiO_2_@PMDA-Si-Tb, SiO_2_@PMDA-Si-Tb-phen with SiO_2_ as the core and terbium organic complex as the shell, were successfully synthesized. The terbium ion was coordinated with organic ligand forming terbium organic complex in the shell layer. The bi-functional organosilane ((HOOC)_2_C_6_H_2_(CONH(CH_2_)_3_Si(OCH_2_CH_3_)_3_)_2_ (abbreviated as PMDA-Si) was used as the first ligand and phen as the second ligand. Furthermore, the silica-modified SiO_2_@PMDA-Si-Tb@SiO_2_ and SiO_2_@PMDA-Si-Tb-phen@SiO_2_ core-shell-shell composites were also synthesized by sol-gel chemical route. An amorphous silica shell was coated around the SiO_2_@PMDA-Si-Tb and SiO_2_@PMDA-Si-Tb-phen core-shell composites. The core-shell and core-shell-shell composites both exhibited excellent luminescence in solid state. The luminescence of core-shell-shell composites was stronger than that of core-shell composites. Meanwhile, an improved luminescence stability property for the core-shell-shell composites was found in the aqueous solution. The core-shell-shell composites exhibited bright luminescence, high stability, long lifetime, and good solubility, which may present potential applications in the bio-medical field.

## 1. Introduction

In the last decade, core-shell composite materials, which connect different functional components that are integrated into one unit, have attracted increased attention. These materials can be used in many fields, including catalysis [1,2,3,4,5,6,7], bio-nanotechnology [8,9,10,11], materials chemistry [12,13], optical devices [14,15,16,17,18], electronics [19,20,21,22], and magnetic devices [23,24,25,26,27]. Among the number of core-shell composite materials [28,29,30,31,32], coating desirable compounds onto a core to form luminescence materials has emerged as one of the research hot spots in recent years. When compared with unary substance, these core-shell composite materials often exhibit improved physical and chemical properties [33,34,35,36], such as good stability, multi-functionality, luminous intensity, fluorescence quantum efficiency, and fluorescence lifetime.

In core-shell composite materials, silicon particles are very important core material because of their aqueous solubility, surface tailorability, low cytotoxicity, and low cost [37,38]. Moreover, the surface of silicon particles contains many active free hydroxyl groups and can be chemically bonded to a substance with a functional property. These core-shell composites not only keep core materials stable, but they also have special shell layer physicochemical properties. So far, there are two commonly used methods to fabricate core-shell composites. One method is direct precipitation, in which compounds can be deposited on the surface of the SiO_2_ core and then form core-shell composites. However, the degree of surface coverage is low and the coating is not uniform. Meanwhile, the prepared composite is unstable and it may easily collapse. Another method is the silane coupling agent method. Silane coupling agent is a bi-functional organics (Y(CH_2_)_n_SiX_3_). Y refers to the organic functional groups, such as amino, carboxyl and double nitrogen. The Y group could coordinate with rare earth ions, forming rare earth organic complexes. The X expresses the alkoxy. The Si–O–Si chemical bonds are constructed after the hydrolysis process of X group and the hydroxyl of SiO_2_ surface. In such a method, the silane coupling agent acted as a “bi-functional molecular bridge”, connecting SiO_2_ spheres and rare earth complexes together. The core-shell composites are stable and the thickness of the shell is easily controlled. Silane coupling agent method is considered to be an effective and popular strategy.

There are sufficient researches on core-shell composite materials with SiO_2_ as the core and inorganic materials as the cladding layer [39,40,41]. However, the inorganic materials usually exhibited weak luminescence property and restricted their practical application. Therefore, in order to obtain excellent luminescence functional materials, we employed the rare earth organic complexes as the coating layer, which can make full use of the "antenna effect" of organic ligand to achieve higher luminous efficiency [42]. When the rare earth organic complex was fixed on the surface of the silica sphere, the non-radiative pathways of high vibration energy loss in the ligand molecules could be largely reduced. In previous work, we have systematically studied the change of luminescence properties of the core-shell composites. Since the thickness of rare earth organic complexes coating layer is nanometer magnitude, it can significantly save the rare earth resources and greatly reduce the production cost [43,44,45,46]. These core-shell composites can exhibit excellent luminescence in the solid state; however, they usually presented poor luminescent stability properties under aqueous medium. Thus, it is necessary to improve the luminescent stability. The silica shell plays an important role on preventing rare earth core-shell composites from quenching of the external environment. In addition, the silica shell can also improve the solubility and luminescent stability of the core-shell composite materials. For example, Ansari et al. prepared hierarchical CePO_4_:Tb@LaPO_4_@SiO_2_ core-shell-shell composites that had significant application in enhancing the solubility, colloidal stability character, and luminescence properties [47]. Therefore, the silica-modified sol-gel technique is an ideal option to improve the solubility and luminous stability of rare earth core-shell composites [48]. Such silica coating core-shell composites simultaneously exhibited multi-functionality [49].

In this work, a silane coupling agent method was presented for the synthesis of SiO_2_@PMDA-Si-Tb and SiO_2_@PMDA-Si-Tb-phen core-shell composites. Specifically, the synthetic approach involved the preparation of organosilane, which acted as a “bi-functional molecular bridge” connecting the silica core and the terbium organic complex shell together by a hydrolysis forming Si–O–Si covalent bonds. Furthermore, the silica-modified SiO_2_@PMDA-Si-Tb@SiO_2_ and SiO_2_@PMDA-Si-Tb-phen@SiO_2_ core-shell-shell composites were prepared using the TEOS-CTAB sol–gel chemical route. In such a strategy, an amorphous silica shell was successfully coated on the surface of SiO_2_@PMDA-Si-Tb and SiO_2_@PMDA-Si-Tb-phen core-shell composites. Photoluminescence measurement showed that the core-shell-shell composites SiO_2_@PMDA-Si-Tb@SiO_2_ and SiO_2_@PMDA-Si-Tb-phen@SiO_2_ can exhibit good luminescence properties and stability both in solid state and in aqueous solution medium. These core-shell-shell composite materials with surface-attached abundant Si–OH molecules are easily available for conjugation with biomolecules. Therefore, they could be used for realistic testing of the biomedical sciences.

## 2. Materials and Methods

### 2.1. Material and Reagents

Tb_4_O_7_ (99.99%), ammonia (25–28%), cetyltrimethylammonium bromide (CTAB, 99%) were all provided by Sigma-Aldrich (Steinheim, Germany). 3-Aminopropyltriethoxysilane (APTES, 99%) was obtained from Acros Organics (Geel, Belgium). Pyromellitic dianhydride (PMDA, 99.2%), 1, 10-phenanthroline (phen, 99%), and Si(OC_2_H_5_)_4_ (TEOS, 99%) were obtained from Sinopharm Chemical Reagent Co., Ltd. (Shanghai, China). All of the reagents were analytical grade and were used as received without further purification. The terbium perchlorate (Tb(ClO_4_)_3_·*n*H_2_O) was prepared by dissolving Tb_4_O_7_ (99.99%) in HClO_4_ (1 mol L^−1^) and then evaporated and dried in vacuum.

### 2.2. Synthesis of Organic Ligand (2,5-bis((3-(Triethoxysilyl)propyl)carbamoyl)terephthalic Acid (PMDA-Si))

PMDA-Si was prepared by following the literature method [50,51]. 1 mmol (0.22 g) pyromellitic dianhydride and 25 mL acetone were placed in a 100 mL three-necked round-bottom flask. After the heterogeneous solution was cooled with an ice bath to 0~5 °C, 2 mmol (0.46 mL) APTES was added dropwise. After completion of the addition of the silane, the solution was continually cooled for 2 h and was then allowed to stand for 10 h with stirring at room temperature. Subsequently, unreacted acetone was distilled off under reduced pressure. The solid residue was further washed with mineral ether and then dried under vacuum at 50 °C to obtain PMDA-Si as a white powder. The synthesis route is shown in Scheme 1. The resulting PMDA-Si was characterized by ^1^H NMR spectrum (Appendix A) and its data is as follows: ^1^H-NMR (*d_6_*-DMSO): δ (ppm): 0.61 (4H)|1.15 (18H)|1.56 (4H)|3.33 (4H)|3.75 (12H)|7.69 (2H)|8.48 (2H)|13.3 (2H); Elemental analysis C_28_H_48_N_2_O_12_Si_2_: Calculated C: 50.89|H: 7.32|N: 4.24; Found C: 50.80|H: 7.43|N: 4.35; Yield: 72.71%.

### 2.3. Synthesis of Silica Cores

The highly monodisperse silica spheres with sizes of around 178 nm were synthesized by the well-known Stöber process [52]. Typically, 1.7 mL TEOS was rapidly dropped into a mixture that consisted of 40 mL anhydrous ethanol, 4.0 mL deionized water, and 1.7 mL ammonia in a 100 mL single-neck round-bottom flask under vigorously stirred solution. The reaction mixture was then placed in an automatic microwave synthesizer (the pressure was 15 PSI, the power was 150 W, the rotational speed was medium speed, and the reaction temperature was 50 °C, respectively). After 4 h of stirring, the resulting white silica suspension was centrifugally separated and then washed with ethanol and deionized water several times after drying at 50 °C in an oven.

### 2.4. Synthesis of SiO_2_@PMDA-Si

The preparation of SiO_2_@PMDA-Si spheres with mass ratio of SiO_2_ to PMDA-Si was 10:13. Briefly, 0.20 g SiO_2_ spheres were ultrasonically dispersed into 20 mL ethanol forming a homogeneous dispersion, followed by the dropwise addition of benzene (30 mL) containing PMDA-Si (0.26 g) under continuous stirring at room temperature. The solution was stirred for 12 h to get a white homogenous mixture. Subsequently, the products were centrifuged, washed thoroughly with anhydrous ethanol successively and repeatedly, and then dried in an oven overnight at 50 °C.

### 2.5. Synthesis of SiO_2_@PMDA-Si-Tb and SiO_2_@PMDA-Si-Tb-phen

Typically, 0.15 g SiO_2_@PMDA-Si spheres were dispersed in 15 mL anhydrous ethanol through ultrasonication. Afterward, 0.07 g Tb(ClO_4_)_3_·*n*H_2_O that was dissolved in 5.0 mL anhydrous ethanol was added dropwise into the above solutions under vigorous stirring. The reaction mixture was heated at 50 °C for 5 h forming a white precipitate. Subsequently, the resulting white precipitate was collected after centrifugation and dried in an oven overnight at 50 °C. The synthetic procedure for SiO_2_@PMDA-Si-Tb-phen was as follows; 0.15 g SiO_2_@PMDA-Si spheres and 0.14 g second ligand phen were dispersed in 15 mL anhydrous ethanol through ultrasonication. Afterwards, 0.20 g Tb(ClO_4_)_3_·*n*H_2_O dissolved in 5.0 mL anhydrous ethanol was slowly added into the above solution under vigorous stirring. The reaction mixture was heated at 50 °C for 5 h. The solution colour changed from white to light pink. Finally, the resulting precipitate was collected by centrifugation and then dried in an oven overnight at 50 °C.

### 2.6. Synthesis of SiO_2_@PMDA-Si-Tb@SiO_2_ and SiO_2_@PMDA-Si-Tb-phen@SiO_2_

The Stöber sol-gel technique was popular for silica surface modification. The specific procedure was as follows, the core-shell composite SiO_2_@PMDA-Si-Tb or SiO_2_@PMDA-Si-Tb-phen (0.03 g), CTAB (0.02 g) and ammonia (0.30 mL) were placed in a 50 mL bottle and well dissolved in a 15 mL mixed solvent of ethanol (7.5 mL) and deionized water (7.5 mL). After stirring the mixture for 10 min at room temperature, 0.03 mL TEOS dissolved in 5.0 mL anhydrous ethanol was added dropwise to the above dispersion to get a white suspension. After continuous stirring for at least 24 h, the final precipitate was separated from the reaction mixture by centrifugation, washed with ethanol, and dried at 50 °C.

### 2.7. Characterization and Apparatus

^1^H-NMR spectroscopy was performed on an AVANCE III 600 M liquid ^1^H-NMR instrument (NMR, Bruker, Karlsruhe, Germany) using DMSO as the solvent and TMS as the internal standard. Elemental analyses were performed with an elemental C, H, and N analyzer. The structure and surface morphology of microspheres were studied with scanning electron microscope (SEM, Hitachi S-4800, Tokyo, Japan) and transmission electron microscope (TEM, FEI Tecnai F20, Hillsboro, OR, USA), accompanied by energy dispersive X-ray spectroscopy (EDX) to examine the chemical composition. X-ray photoelectron spectroscopy (XPS) was utilized to study surface chemical analysis by Thermo ESCALAB 250Xi spectrometer (Thermo Fisher Scientific, Waltham, MA, USA) with an Al kα radiation source. Thermogravimetric analysis (TGA) was performed on Netzsch STA 409 thermogravimetric analyser (TA Instrument, Selb, Germany) at a heating rate of 20 °C min^−1^ under an atmosphere of nitrogen gas. Fourier transform infrared spectra (FT-IR) were measured on a VERTEX 70 spectrometer (FT-IR, Bruker, Karlsruhe, Germany) at 500–4000 cm^−1^ using the KBr pellet method. X-ray powder diffraction (XRD, RIGAKU, Tokyo, Japan) was measured by a 21 kW extra power X-ray diffractometer using Cu Kα radiation (λ = 1.5406 Å), over an angle range from 5–80°. The photoluminescence spectra and the quantum yields of the solid-state products were measured on a photoluminescence spectrometer (FL; Edinburgh S980, Livingston, UK). The phosphorescence spectra of powder samples were monitored using a FLS980 spectrophotometer at 77 K (FL; Edinburgh S980, Livingston, UK).

### 2.8. Lifetime Analysis

The photoluminescence lifetime analysis for the representative emission of Tb^3+^ ions was performed using an Edinburgh FLS980 spectrophotomete at room temperature. The decay curve can be fitted into a biexponential function (1). The lifetime values of the excited state terbium ion (^5^D_4_) could be calculated using the following Equation (2), where τ_1_ and τ_2_ stand for the slow and fast terms of the luminescent lifetime, respectively, and A_1_ and A_2_ are the corresponding pre-exponential factors. 

I_(t)_ = I_0_ + A_1_ exp(−t_1_/τ_1_) + A_2_ exp(−t_2_/τ_2_)(1)

(τ) = (A_1_τ_1_^2^ + A_2_τ_2_^2^)/(A_1_τ_1_ + A_2_τ_2_)(2)

## 3. Results and Discussion

### 3.1. The Formation Mechanism and Design Idea of Core-Shell and Core-Shell-Shell Composites

The synthesis strategy for fabricating core–shell and core-shell-shell composites is presented in Scheme 2. It is a key step in synthesizing the high surface functionality of silica particles with molecular bridge PMDA-Si. First, the synthetic approach involved the preparation of silica sphere and organosilane PMDA-Si. The PMDA-Si acted as a “bi-functional molecular bridge” and the alkoxyl groups can be covalently bonded with the active hydroxyl groups on the surface of silica spheres forming Si–O–Si bonds. In this process, organosilane PMDA-Si was successfully grafted on the surface of silica sphere. Thus, the carboxyl group of PMDA-Si could coordinate with terbium ions. Additionally, the introduction of the second ligand phen could also synergistically coordinate to terbium ions by double nitrogen atoms and form a terbium organic complex in the shell layer. Here, the core-shell composites were synthesized with SiO_2_ as the core and terbium organic complex as the shell. To enhance the solubility and the luminescent stability of the core-shell composite materials, the silica-modified core-shell-shell composites were synthesized. An amorphous silica shell was uniformly coated around these as-prepared core-shell composites by the sol-gel chemical route, which involved the hydrolysis and polycondensation process of TEOS while using CTAB as a surface-active agent. As an intermediate layer, the terbium organic complexes were fixed on the inorganic silica substrate. The non-radiative pathways of high vibration energy loss in the ligand molecules could be largely reduced. As a result, it was expected that the emission efficiency of core-shell-shell composites were significantly enhanced. A higher luminescent stability will be obtained under aqueous medium, coming from the protection of an amorphous silica shell.

### 3.2. Morphology and Structure

#### 3.2.1. The TEM of SiO_2_@PMDA-Si-Tb and SiO_2_@PMDA-Si-Tb-phen

The morphological structure and size of the as-prepared SiO_2_ are shown in Figure 1a,b. Based on the SEM and TEM micrograph, SiO_2_ has a spherical shape and a smooth surface with a mean diameter of around 178 nm. When the functional molecular bridge PMDA-Si was grafted onto the surface of SiO_2_, as can be seen with a low-magnification SEM image (Figure 1d), these SiO_2_@PMDA-Si particles exhibited perfect uniformity and monodispersity. The high-magnification TEM image (Figure 1e) indicates that the SiO_2_@PMDA-Si particles possess a relatively rough surface with a mean diameter of 186 nm and thickness of 4 nm. The corresponding histograms of the size distribution are presented in Figure 1c,f.

Simultaneously, the formation of SiO_2_@PMDA-Si-Tb and SiO_2_@PMDA-Si-Tb-phen core-shell composites was clearly verified by SEM and TEM. When compared with SiO_2_@PMDA-Si particles (Figure 1d), the surface morphologies of SiO_2_@PMDA-Si-Tb and SiO_2_@PMDA-Si-Tb-phen (Figure 2a,d) differed from SiO_2_@PMDA-Si sphere. After being coated with terbium organic complex layer, the surface of SiO_2_@PMDA-Si-Tb and SiO_2_@PMDA-Si-Tb-phen core-shell composites became a little larger and much rougher. Figure 2b,c,e,f show the TEM images at the SiO_2_@PMDA-Si-Tb and SiO_2_@PMDA-Si-Tb-phen core-shell composites, respectively. These core-shell composites were still spherical and nonaggregated, but a slight increase in diameter than the SiO_2_@PMDA-Si particles. The changes in diameter were from 178 nm for SiO_2_ spheres to 190 nm for SiO_2_@PMDA-Si-Tb and SiO_2_@PMDA-Si-Tb-phen. It indicated that the terbium organic complexes were coated on the surface of SiO_2_ spheres with the thicknesses of about 6 nm. In addition, the element composition of the SiO_2_@PMDA-Si-Tb and SiO_2_@PMDA-Si-Tb-phen core-shell composites was subsequently analyzed by energy dispersive X-ray spectroscopy (EDX) in Figure 3g, which suggested the existence of Si, O, N, Cl, and Tb elements in the core-shell composites. These results seem to indicate that the SiO_2_@PMDA-Si-Tb and SiO_2_@PMDA-Si-Tb-phen core-shell composites were successfully synthesized.

#### 3.2.2. The TEM of SiO_2_@PMDA-Si-Tb@SiO_2_ and SiO_2_@PMDA-Si-Tb-phen@SiO_2_

In order to widen the application of core-shell composite materials in bio-medical fields, the silica-modified SiO_2_@PMDA-Si-Tb@SiO_2_ and SiO_2_@PMDA-Si-Tb-phen@SiO_2_ core-shell-shell composites were also synthesized. Further information about the shell growth was provided by TEM analysis. Figure 3a–f shows the TEM images of SiO_2_@PMDA-Si-Tb@SiO_2_ and SiO_2_@PMDA-Si-Tb-phen@SiO_2_. There was a clear morphological difference between the SiO_2_@PMDA-Si-Tb, SiO_2_@PMDA-Si-Tb-phen core-shell composites and the silica-modified SiO_2_@PMDA-Si-Tb@SiO_2_ (Figure 3a–c), SiO_2_@PMDA-Si-Tb-phen@SiO_2_ (Figure 3d–f) core-shell-shell composites. Wherein, it showed that the core-shell composites were uniformly encapsulated into a gray shell, and the diameter increased to approximately 210 nm corresponding to a 10 nm thick layer on the surface assigned to amorphous silica. Moreover, the EDX analysis that was performed on the core-shell-shell composites suggested that the Si, O, N, Cl, and Tb elements peaks in the spectrum still existed, however, the Tb content decreased from 15.54% for SiO_2_@PMDA-Si-Tb (Figure 3g) to 8.22% for SiO_2_@PMDA-Si-Tb@SiO_2_ (Figure 3h). Owing to an amorphous silica layer that was coated onto the surface of SiO_2_@PMDA-Si-Tb and SiO_2_@PMDA-Si-Tb-phen core-shell composites, eventually causing the reduction of Tb content when compared with the corresponding core-shell composites. To further confirm the formation of core-shell-shell composite SiO_2_@PMDA-Si-Tb@SiO_2_, the X-ray elemental mappings analysis was performed. It was clearly showed in the STEM image (Figure 4a) and STEM-EDX elemental mappings (Figure 4b–f) that the elements of N, O, Si, Cl, and Tb were uniformly distributed in the spheres-like area. All of the above results indicated the successful preparation of SiO_2_@PMDA-Si-Tb@SiO_2_ and SiO_2_@PMDA-Si-Tb-phen@SiO_2_ core-shell-shell composites.

### 3.3. XPS Analysis

To further explain the incorporation of Tb-(PMDA-Si)-ClO_4_ and Tb-(PMDA-Si)-phen-ClO_4_ complexes onto the surface of SiO_2_, the XPS spectra of PMDA-Si, SiO_2_@PMDA-Si-Tb and SiO_2_@PMDA-Si-Tb-phen were obtained. As Figure 5a shows, two new peaks at 1240.9 eV and 1274.7 eV appeared after the treatment by Tb^3+^ and then the existence of Tb^3+^ in the composites can be ascertained. In addition, the corresponding binding energy data for Tb was displayed in Figure 5b. The peaks at around 1240.9 eV and 1274.7 eV indicate the binding energy of Tb 3d_5/2_ and 3d_3/2_ [53]. On the basis of above analysis, the peaks corresponding to Tb confirmed the successful incorporation of Tb-(PMDA-Si)-ClO_4_ and Tb-(PMDA-Si)-phen-ClO_4_ complexes onto the surface of SiO_2_.

### 3.4. Thermogravimetric Analysis

The thermal decomposition performance of the synthesized core-shell composite SiO_2_@PMDA-Si-Tb-phen and core-shell-shell composite SiO_2_@PMDA-Si-Tb-phen@SiO_2_ was carried out from ambient temperature to 900 °C under the nitrogen atmosphere. As seen in Figure 6, a smaller weight loss in TGA curve exhibits about 2% and 3% for the core-shell composite and core-shell-shell composite, respectively, which took place in between 28 and 138 °C. This may be attributed to the evaporation of the surface absorbed water and organic solvents from the sample. Furthermore, the thermal decomposition temperature of the core-shell composite SiO_2_@PMDA-Si-Tb-phen (Figure 6a) was 160 °C, while it was increased to 200 °C after coating an amorphous silica layer. The thermal stability of SiO_2_@PMDA-Si-Tb-phen@SiO_2_ composite (Figure 6b) increases a little due to a growth of silica layer on the surface of SiO_2_@PMDA-Si-Tb-phen. A larger weight loss of about 40% and 18% can be seen for the SiO_2_@PMDA-Si-Tb-phen and SiO_2_@PMDA-Si-Tb-phen@SiO_2_ under 650 °C. It can be concluded that the core-shell-shell composite displays higher thermal decomposition temperature and the core-shell-shell composite has better thermal stability.

### 3.5. Infrared Spectra Analysis

#### 3.5.1. The FT-IR Spectra of SiO_2_@PMDA-Si-Tb and SiO_2_@PMDA-Si-Tb@SiO_2_

Further, to verify the purity and internal structure of core-shell SiO_2_@PMDA-Si-Tb and core-shell-shell SiO_2_@PMDA-Si-Tb@SiO_2_, the FT-IR spectra of PMDA-Si, SiO_2_, SiO_2_@PMDA-Si, SiO_2_@PMDA-Si-Tb, and SiO_2_@PMDA-Si-Tb@SiO_2_ are investigated, as shown in Appendix A. The appearance of the characteristic bands of PMDA-Si (Appendix A) at 1633 cm^−1^ and 1550 cm^−1^ was attributed to the stretching vibration of the –CONH– bond, and the stretching vibration of –C=O– (COOH) appeared at 1727 cm^−1^. These characteristic bands demonstrated that functional molecular bridge PMDA-Si had been successfully synthesized through the amidation reaction with pyromellitic dianhydride and 3-aminopropyltriethoxysilane. The band of SiO_2_ (Appendix A) at 1098 cm^−1^ was attributed to the stretching vibration of the Si–O–Si group, and Si–OH group was identified at 952 cm^−1^. In the spectra of SiO_2_@PMDA-Si (Appendix A), the appearance of the characteristic band at 1718 cm^−1^ was ascribed as the stretching vibration of –C=O– (COOH). The characteristic bands at 1641 cm^−1^ and 1556 cm^−1^ originated from the stretching vibration of –CONH–, which further confirmed the "functional molecular bridge" PMDA-Si onto the surface of SiO_2_. The FT-IR spectrum of SiO_2_@PMDA-Si-Tb core-shell composites (Appendix A) shows the characteristic bands of –C=O– (COOH) bonds at 1714 cm^−1^. There was an obvious change in the absorption band when compared with SiO_2_@PMDA-Si, indicating that Tb^3+^ ions coordinated to the oxygen atoms of carbonyl group. In addition, the observed characteristic infrared bands at 1147, 925, and 622 cm^−1^ were assigned to the stretching and bending vibration modes of the perchlorate (ClO_4_^-^) group. As can be observed in the FT-IR spectrum of the SiO_2_@PMDA-Si-Tb@SiO_2_ core-shell-shell composites (Appendix A), there was a significant change between the spectra of SiO_2_@PMDA-Si-Tb and SiO_2_@PMDA-Si-Tb@SiO_2_. After coating the SiO_2_ shell, the intensity of the characteristics of the –CONH– and –C=O– (COOH) absorption bands were obviously decreased. The surface Si–OH groups play an important role for the functionalization and their affinity with bio-molecules, which is usable in the bio-medical field.

#### 3.5.2. The FT-IR Spectra of SiO_2_@PMDA-Si-Tb-phen and SiO_2_@PMDA-Si-Tb-phen@SiO_2_

After further encapsulation by the second ligand phen and SiO_2_ shell, the formation process of the core-shell and core-shell-shell composites was also investigated by FT-IR spectra. Appendix A shows the FT-IR spectra of phen, SiO_2_@PMDA-Si-Tb-phen, and SiO_2_@PMDA-Si-Tb-phen@SiO_2_. When comparing the spectra SiO_2_@PMDA-Si-Tb-phen (Appendix A) with SiO_2_@PMDA-Si (Appendix A), the stretching vibration of –C=O– (COOH) red-shifted to 1714 cm^−1^ and the stretching vibration of –CONH– red-shifted to 1632 cm^−1^ and 1521 cm^−1^, implying that the -C=O- (COOH) of PMDA-Si was coordinated with Tb^3+^. Moreover, the stretching vibration of –C=N– in the spectra of free phen (Appendix A) was located at about 1587 cm^−1^, however, which had shifted to lower frequencies at 1579 cm^−1^ in the spectra of SiO_2_@PMDA-Si-Tb-phen, suggesting that the Tb^3+^ ion coordinated with double nitrogen atoms of phen and Tb-(PMDA-Si)-phen-ClO_4_ complexes that were successfully coated on the surface of the SiO_2_ core. In the FT-IR spectrum of SiO_2_@PMDA-Si-Tb-phen@SiO_2_ (Appendix A), similarly, the intensity of characteristic absorption bands –CONH– and –C=O– (COOH) bonds were obviously decreased, which was deeply indicated that the SiO_2_ shell was effectively encapsulated around the surface of SiO_2_@PMDA-Si-Tb-phen core-shell composites.

### 3.6. X-ray Diffraction Analysis

The composition and structure of the powder samples were examined by XRD. Figure 7A shows the XRD patterns of the SiO_2_ core, SiO_2_@PMDA-Si, SiO_2_@PMDA-Si-Tb, and SiO_2_@PMDA-Si-Tb@SiO_2_, respectively. For SiO_2_ that was directly formed from the Stöber method (Figure 7Aa), there were two broad bands centered at 2*θ* = 7–8° and 2*θ* = 23°, which were identical with the standard XRD pattern for amorphous SiO_2_. After the bi-functional organosilane PMDA-Si was grafted onto the surface of SiO_2_, no other phase was detected from the comparison of SiO_2_ in Figure 7Ab. For SiO_2_@PMDA-Si-Tb core-shell composite in Figure 7Ac, the diffraction pattern also consisted of two broad bands that appeared at 2*θ* = 7–8° and 2*θ* = 23°, and the intensity of the diffraction band became weaker due to the coating of Tb-(PMDA-Si)-ClO_4_ complexes. Besides, there were several sharp peaks at about 7.3°, 8.3°, 13.8°, 20.7°, 22.1°, and 23.8°, indicating that Tb-(PMDA-Si)-ClO_4_ complexes coated on the surface of the SiO_2_ core. As illustrated in Figure 7Ad, the two broad bands of SiO_2_@PMDA-Si-Tb@SiO_2_ core-shell-shell composites appeared at 2*θ* = 7–8° and 23° were in coincidence with the stander XRD pattern for SiO_2_ core (Figure 7Aa), which further illustrated that an amorphous SiO_2_ was coated onto the surface of SiO_2_@PMDA-Si-Tb.

After the incorporation of the phen and SiO_2_ shell, the growth of Tb-(PMDA-Si)-phen-ClO_4_ and SiO_2_ shell onto the core-shell and core-shell-shell composites has been characterized by XRD. Figure 7B shows the XRD patterns of the SiO_2_ core, SiO_2_@PMDA-Si, SiO_2_@PMDA-Si-Tb-phen and SiO_2_@PMDA-Si-Tb-phen@SiO_2_. When compared with the diffraction pattern of SiO_2_ core (Figure 7Ba), several small sharp diffraction peaks at about 7.9°, 8.9°, 12.4°, 18.7°, 21.1°, 23.8°, and 27° were observed for the SiO_2_@PMDA-Si-Tb-phen composites (Figure 7Bc). It means the existence of Tb-(PMDA-Si)-phen-ClO_4_ on the surface of the SiO_2_ core. The diffraction pattern of the SiO_2_@PMDA-Si-Tb-phen@SiO_2_ composites (Figure 7Bd) displayed the disappearance of several small diffraction peaks. Moreover, the broad bands that were centered at 2*θ* = 23° shifted from little to larger diffraction angles when compared with the pure SiO_2_ core. This phenomenon may be attributed to the growth environment of the silica core and the silica shell was different.

### 3.7. The Photoluminescence Properties

To evaluate the effect of surface modification of SiO_2_ on optical absorption properties, photoluminescence spectra were performed to get more information regarding the optical behavior of the Tb(III) core-shell and core-shell-shell composites. As shown in Figure 8, the room temperature fluorescent excitation and emission spectra of core-shell composites SiO_2_@PMDA-Si-Tb, SiO_2_@PMDA-Si-Tb-phen and core-shell-shell composites SiO_2_@PMDA-Si-Tb@SiO_2_, SiO_2_@PMDA-Si-Tb-phen@SiO_2_ were investigated in solid state with the slit width of 1.0 nm.

The excitation spectra that were monitored at 543 nm were investigated in Figure 8a,c. It can be clearly observed that all excitation spectra exhibited a very broad band in the region of 200–400 nm. The corresponding emission spectra were also measured in Figure 8b,d. These sharp emission peaks at about 488 nm, 543 nm, 583 nm, and 621 nm were related to ^5^D_4_→^7^F_6_, ^5^D_4_→^7^F_5_, ^5^D_4_→^7^F_4_, and ^5^D_4_→^7^F_3_ transitions of terbium ions, while the ^5^D_4_→^7^F_5_ transition at about 543 nm was the strongest [54,55,56]. Table 1 displays the photoluminescence emission spectra data of the Tb(III) core-shell and core-shell-shell composites. The core-shell composites and core-shell-shell composites both exhibited a much stronger luminescent properties in solid state. Particularly, the core-shell-shell composites displayed stronger luminescence. The strongest emission intensity of SiO_2_@PMDA-Si-Tb@SiO_2_ and SiO_2_@PMDA-Si-Tb-phen@SiO_2_ was 7,420,000 a.u. and 7,314,118 a.u., respectively. When compared with the corresponding terbium core-shell composites, the emission intensity of core-shell-shell composites SiO_2_@PMDA-Si-Tb@SiO_2_ and SiO_2_@PMDA-Si-Tb-phen@SiO_2_ were about 3.64 and 1.96 times higher. Furthermore, the photoluminescence properties of core-shell and core-shell-shell composites were compared to those of inorganic luminescent material. It is interesting to note that the emission intensities in core-shell and core-shell-shell composites were relatively stronger under the same measurement conditions [57,58]. To further investigate the photoluminescence properties of Tb(III) core-shell and core-shell-shell composites, the absolute quantum yield was determined. The absolute quantum yields of SiO_2_@PMDA-Si-Tb and SiO_2_@PMDA-Si-Tb-phen were 33.18% and 36.57%, respectively. In contrast, the corresponding absolute quantum yields of SiO_2_@PMDA-Si-Tb@SiO_2_ and SiO_2_@PMDA-Si-Tb-phen@SiO_2_ were 41.75% and 47.34%, respectively. When encapsulated with an amorphous silica shell, the core-shell-shell composite materials have a marked increase in absolute quantum yield. These results may be attributed to the fact that non-radiative centers that exist on the surface of core-shell composites were eliminated by the shielding effect of the silica shell. It may also result from the suppression of the OH groups quenching effect by the silica shell. Meanwhile, the emission intensities of core-shell-shell composites were significantly enhanced after the coating of an amorphous silica shell. The introduction of the second ligand phen can also synergistically sensitize the luminescence of terbium ions by the "antenna effect" and then largely improve the luminescent properties of the core-shell composites.

In this work, the luminescence stability of the core-shell composites and core-shell-shell composites in aqueous solution was also studied. The SiO_2_@PMDA-Si-Tb, SiO_2_@PMDA-Si-Tb-phen and SiO_2_@PMDA-Si-Tb@SiO_2_, SiO_2_@PMDA-Si-Tb-phen@SiO_2_ composites were dissolved in deionized water at a concentration of 0.1 g L^−1^, and the change of luminescence properties after being placed 0, 6, 12, 24, 48 and 72 h, with the slit width of 2.0 nm, were investigated. The aqueous solution was sonicated for 10 min before each measurement to achieve homogeneous suspension. Figure 9a–d displayed the room temperature fluorescent emission spectra of the composites. It was clearly demonstrated that both the emission spectra of the core–shell and core-shell-shell composites displayed four typical peaks at about 488, 543, 583, and 621 nm, which were attributed to ^5^D_4_→^7^F_6_, ^5^D_4_→^7^F_5_, ^5^D_4_→^7^F_4_, and ^5^D_4_→^7^F_3_ transitions of terbium ions. These results corresponded to the previous study. In addition, with the time increasing from 0 h to 72 h, the decrease degree of emission intensity for core-shell-shell composites were obviously smaller than that for core-shell composites. The core-shell-shell composites exhibited better luminescence stability in the aqueous medium. Especially, the core-shell-shell structure was not collapsed, even after three days. The introduction of an amorphous silica shell not only can suppress OH groups quenching effect from the surrounding environment, but it can also protect the internal structure of the core-shell composites. Therefore, a better luminescent stability in aqueous solution was obtained. These silica-modified core-shell-shell composites, which have been proved to enhance the luminescence efficiency, should be favorable for potential biological application.

### 3.8. The Photoluminescence Lifetime

In order to study the details of photoluminescence property for terbium core-shell and core-shell-shell composites, the photoluminescence decay curve of SiO_2_@PMDA-Si-Tb, SiO_2_@PMDA-Si-Tb-phen and SiO_2_@PMDA-Si-Tb@SiO_2_, SiO_2_@PMDA-Si-Tb-phen@SiO_2_ composites was recorded in Figure 10. The decay curve can be well fitted into a biexponential function. Herein, the calculated average lifetimes of two terbium core-shell composites SiO_2_@PMDA-Si-Tb and SiO_2_@PMDA-Si-Tb-phen were 927.32 μs and 960.85 μs, respectively. Moreover, the average lifetime of two terbium core-shell-shell composites SiO_2_@PMDA-Si-Tb@SiO_2_ and SiO_2_@PMDA-Si-Tb-phen@SiO_2_ were 1086.28 μs and 970.18 μs, respectively. The increase of the photoluminescence lifetime for two terbium core-shell-shell composites showed that the quenching from outside particles was strongly reduced after the growth of a silica shell around the core-shell composites. The detail values are listed in Table 2. Two terbium core-shell-shell composites both have relatively long lifetimes under UV excitation.

### 3.9. Low-temperature Phosphorescence Spectra

To verify the intra-molecular energy transfer from the triplet state of the organic ligand to the resonance level of rare earth ion, the phosphorescence spectrum of the first ligand PMDA-Si and second ligand phen was measured under the irradiation of a 327 nm and 372 nm UV lamp, with slit widths of 1.0 nm and 10.0 nm in the solid state at 77 K (Appendix A). There were two relatively symmetric emission bands between 446 nm and 520 nm in the phosphorescence spectrum of PMDA-Si, as shown in Appendix A. The first band ranged from 455 nm to 520 nm, which was defined as T_1_. The second band ranged from 446 nm to 454 nm, which was defined as T_2_. The triplet state energy level of T_1_ was 21978–19231 cm^−1^ and the triplet state energy level of T_2_ was 22422–22026 cm^−1^. These results suggested that the triplet state energy level of the first ligand PMDA-Si was higher than ^5^D_4_ of Tb(III) ions (20430 cm^−1^) (Appendix A). Therefore, the first ligand PMDA-Si could effectively transfer the energy absorbed in the ultraviolet region to the central terbium ions by non-radiative transition mode, sensitizing the luminescence of terbium ions. The SiO_2_@PMDA-Si-Tb core-shell composites had excellent luminescence.

The phosphorescence spectrum of phen is shown in Appendix A, where two phosphorescence emission bands were reflected. The first band ranged from 502 nm to 622 nm, which was defined as T_1_. The triplet state energy level of T_1_ was 19920–16077 cm^−1^. The second band ranged from 440 nm to 475 nm, which was defined as T_2_. The triplet state energy level of T_2_ was 22727–21053 cm^−1^. It was also illustrated that the triplet state energy level of the second ligand phen and the excited state energy of terbium ions matched very well (Appendix A), and that the second ligand could effectively transfer the energy that was absorbed in the ultraviolet region to the central terbium ions and sensitize the luminescence of terbium ions. Therefore, according to the low-temperature phosphorescence spectra studies, the organic ligands of PMDA-Si and phen had a synergistic reaction to transfer the energy absorbed in the ultraviolet region to the central terbium ions. The photoluminescence intensity of SiO_2_@PMDA-Si-Tb-phen was much stronger than the SiO_2_@PMDA-Si-Tb core-shell composites.

## 4. Conclusions

Two novel spherical core-shell composites SiO_2_@PMDA-Si-Tb and SiO_2_@PMDA-Si-Tb-phen were successfully synthesized with organosilane PMDA-Si acting as a “bi-functional molecular bridge”. Moreover, silica-modified SiO_2_@PMDA-Si-Tb@SiO_2_ and SiO_2_@PMDA-Si-Tb-phen@SiO_2_ core-shell-shell composites were also successfully synthesized. The formation mechanisms of the core-shell and core-shell-shell composite were proposed. The TEM and STEM images showed that the terbium organic complex was successfully coated on the surface of SiO_2_ spheres and the amorphous silica shell was uniformly encapsulated around these as-prepared core-shell composites. The core-shell and core-shell-shell composites both exhibited much stronger luminescent properties in solid state. The photoluminescence emission intensity of SiO_2_@PMDA-Si-Tb@SiO_2_ and SiO_2_@PMDA-Si-Tb-phen@SiO_2_ was about 3.64 and 1.96 times higher when compared with the homologous terbium core-shell composites. By coating a silica shell on the surface of core-shell composites, better luminescence properties and stability for SiO_2_@PMDA-Si-Tb@SiO_2_ and SiO_2_@PMDA-Si-Tb-phen@SiO_2_ were found in the aqueous medium. In the core-shell-shell composite materials, the cheap silica was used as supporting cores and the nanometer magnitude terbium organic complex was employed as coating layers, which could significantly save rare earth resources and reduce the production cost. In sum, the whole synthesis procedure of the core-shell-shell composites was quite facile and easy to scale up, which holds great potential for low-cost commercial applications.

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
