# Peer review of "Synthesis and Luminescence Properties of Core-Shell-Shell Composites: SiO2@[email protected]2 and SiO2@[email protected]2"

_nanomaterials, 2019, doi:10.3390/nano9020189_

Reviewer 1 Report

This work is quite interesting and I think it can be considered for publication in Nanomaterials. 

I am a bit concerned that the manuscript is lacking results of TGA. How temperature stable are these materials? Also what is the stability of the materials after dispersing in water? DO they deteriorate after days, weeks? Has this been tested. 

Another remark is that in the SI in Figure S4 its seems to me that the sam results are presented as in Figure 10 in the manuscript. If the decay curves were meant to be presented in the SI please change y scale to logarithmic and add the fitting to the decay curves. 

Author Response

Point 1: This work is quite interesting and I think it can be considered for publication in Nanomaterials. I am a bit concerned that the manuscript is lacking results of TGA. How temperature stable are these materials? Also what is the stability of the materials after dispersing in water? DO they deteriorate after days, weeks? Has this been tested.

Response 1: We are very grateful for the reviewer’s suggestion. The thermogravimetric analysis had been measured. The thermal decomposition temperature of core-shell composite SiO2@PMDA-Si-Tb-phen is 160 °C, while it is increased to 200 °C after coating an amorphous silica layer. The core-shell-shell composite displays higher thermal decomposition temperature and has better thermal stability. 

(Page 8, Line 276-292)

3.4. Thermogravimetric Analysis

The thermal decomposition performance of the synthesized core-shell composite SiO2@PMDA-Si-Tb-phen and core-shell-shell composite SiO2@[email protected]2 was carried out from ambient temperature to 900 °C under nitrogen atmosphere. As seen in Figure 7, a smaller weight loss in TGA curve exhibits about 2% and 3% for core-shell composite and core-shell-shell composite, respectively, which took place in between 28 and 138 °C. This may be attributed to the evaporation of the surface absorbed water and organic solvents from the sample. Furthermore, the thermal decomposition temperature of core-shell composite SiO2@PMDA-Si-Tb-phen (Figure 7a) was 160 °C, while it was increased to 200 °C after coating an amorphous silica layer. The thermal stability of SiO2@[email protected]2 composite (Figure 7b) increases a little due to a growth of silica layer on the surface of SiO2@PMDA-Si-Tb-phen. A larger weight loss of about 40% and 18% can be seen for the SiO2@PMDA-Si-Tb-phen and SiO2@[email protected]2 under 650 °C. It can be concluded that the core-shell-shell composite displays higher thermal decomposition temperature, and the core-shell-shell composite has better thermal stability. 

Figure 7. Thermogravimetric analysis curves of the SiO2@PMDA-Si-Tb-phen (a) and SiO2@[email protected]2 (b).

We are very grateful for the reviewer’s suggestion. Further to investigate the luminescence stability of the core-shell composites and core-shell-shell composites in aqueous solution, SiO2@PMDA-Si-Tb, SiO2@PMDA-Si-Tb-phen and SiO2@[email protected]2, SiO2@[email protected]2 composites were dissolved in deionized water at a concentration of 0.1 g L-1 and were investigated the change of luminescence properties after placed 0 h, 6 h, 12 h, 24 h, 48 h, 72 h with the slit width of 2.0 nm. The luminescent measurement illustrated that with the increase of time from 0 h to 72 h, the decrease degree of emission intensity for core-shell-shell composites was obviously smaller than of the core-shell composites. The core-shell-shell composites exhibited better luminescence stability in the aqueous medium and the core-shell-shell structure was not collapsed after three days. 

(Page 12-13, Line 405-427)

In this work, the luminescence stability of the core-shell composites and core-shell-shell composites in aqueous solution was also studied. The SiO2@PMDA-Si-Tb, SiO2@PMDA-Si-Tb-phen and SiO2@[email protected]2, SiO2@[email protected]2 composites were dissolved in deionized water at a concentration of 0.1 g L-1, and were investigated the change of luminescence properties after placed 0, 6, 12, 24, 48 and 72 h with the slit width of 2.0 nm. The aqueous solution was sonicated for 10 min before each measurement to achieve a homogeneous suspension. Figure 10a-d displayed the room temperature fluorescent emission spectra of the composites. It was clearly demonstrated that both the emission spectra of the core–shell and core-shell-shell composites shown four typical peaks at about 488, 543, 583 and 621 nm, which attributed to 5D47F6, 5D47F5, 5D47F4 and 5D47F3 transitions of terbium ions. These results corresponded to the previous study. In addition, with the time increasing from 0 h to 72 h, the decrease degree of emission intensity for core-shell-shell composites were obviously smaller than that for core-shell composites. The core-shell-shell composites exhibited better luminescence stability in the aqueous medium. Especially, the core-shell-shell structure was not collapsed even after three days. The introduction of an amorphous silica shell not only can suppress OH groups quenching effect from the surrounding environment but also can protect the internal structure of the core-shell composites. Therefore, a better luminescent stability in aqueous solution was obtained. This silica-modified core-shell-shell composites, which have been proved to enhance the luminescence efficiency, should be favorable for potential biological application. 

Figure 10. Fluorescent emission spectra of SiO2@PMDA-Si-Tb (a), SiO2@PMDA-Si-Tb-phen (b), SiO2@PMDA-Si-Tb@SiO2 (c) and SiO2@PMDA-Si-Tb-phen@SiO2 (d) after placement for 0 h, 6 h, 12 h, 24 h, 48 h and 72 h in aqueous solution.

Point 2: Another remark is that in the SI in Figure S4 its seems to me that the same results are presented as in Figure 10 in the manuscript. If the decay curves were meant to be presented in the SI please change y scale to logarithmic and add the fitting to the decay curves.

Response 2: Thanks for the reviewer’s carefully reading and great suggestion. Figure S4 have been deleted in the revised manuscript. 

Reviewer 2 Report

Major comment 

At present, the field of the organic/inorganic hybrids with silica core and lanthanide organic complexes is quite inflated. The authors should correctly place their findings in the literature context, highlighting the novelty and applications. As the luminescence is apparently the proposed functionality of these compounds, the authors should compare it with the performance of similar compounds reported so far. To this aim, estimation of the quantum yield is mandatory. Otherwise, the term brightness is not supported by any of presented data. In fact, as seen from the Table S1, the variation in the average lifetimes is not significant and can be assigned mainly to fitting procedure. This make the whole synthesis useless for any possible applications compared to what is published already.

Other comments 

Synthesis and characterization

Page 3 row 126: “For the preparation of [email protected] spheres, briefly, 0.20 g SiO2 spheres were ultrasonically dispersed into 20 mL ethanol forming a homogeneous dispersion, followed by dropwise addition of benzene (30 mL) containing PMDA-Si (0.26g) under continuous stirring at room temperature” Authors may need to provide additional details about the ratio between SiO2 spheres and PMDA-Si. Details such as the molar ratio alongside the weight ratio and the use or not of excess of a component may help the readability of the text.

Page 3 row 136: “The synthetic procedure for [email protected] was as follows, 0.15 g [email protected] spheres and 0.14 g second ligand phen were dispersed in 15 mL anhydrous ethanol under continuous stirring. Then 0.20 g Tb(ClO4)3· nH2O dissolved in 5.0 mL anhydrous ethanol was added, and the mixture was heated at 50 °C for 5 h”

Please confirm that you have added phen over the SiO2@PMDA-Si and afterwards Tb on top of that mixture.

Page 4 row 169:” The PMDA-Si acted as a "bi-functional bridge molecular” ”Word inversion of the quoted term as “bi-functional molecular bridge”.

Row 170: “connecting (…) by a hydrolysis forming Si-O-Si covalent bonds”. This sentence is ambiguous and authors should consider to revise this part.

Row 175: “In this step, because silica as supporting cores was cheap and terbium organic complex as coating layer was nanometer magnitude, the core-shell composites can significantly save rare earth resources and greatly reduce production cost” Please, detail what is the discussed step. Also, the introduction of the cost benefits in the Formation Mechanism and Design Idea is diverging from the aim of this paragraph. Please, reassign this sentence to other chapter or to place it in a different section.

Row 178: “Furthermore, as one end of the terbium organic complexes were fixed on the surface of SiO2 core, the vibration of the ligand molecules (PMDA-Si, phen) could be largely reduced” A clarification is needed in this sentence due to the appearance for the first time of “terbium organic complexes”. Row 179: “The core-shell composites exhibited excellent luminescence properties”. This sentence is misplaced and should be relocated.

Row 181: “The other end was still in contact with the external environment, which had a quenching effect on the rare earth organic complex shell” Confuse statement, please revise it.  Page 5 row 187: “inorganic substrate silica”  Word inversion.

Row 190: “excitation efficiency enhancement” What do the author mean by this statement ?

The description of TEM data and core-shell structure should be strongly revised. Please explain use of terms such as “dark centers” and “pale edges” for describing the core-shell structures and should provide a scientific explanation regarding those dark and pale regions.

Row 222: The statement that the terbium complexes were successfully synthetized is expressed twice in the same paragraph.

Row 230: “In order to widen the application field of core-shell composite materials, the silica-modified core-shell-shell composites were also synthesized” What applications have the authors in mind ?

Row 236: “Wherein, it showed the black core-shell composites were uniformly encapsulated into a thin gray silica shell” Please, clarify this paragraph.  

Row 260: “We characterized the XPS spectra” Use interpret or analyse instead of characterization. Further, XPS provides information about the surface chemistry and not whole sample chemistry and in consequence the 3.3 Chemical analysis title is not appropriate.

Section  3.4.1: The successful synthesis of those particles is stated three times in this section and can be a conclusion of this part. The authors should rephrase the section.

Section 3.5 : The authors note the 2Theta values of the diffraction peaks but do not mention the observed shift towards higher angles between the two samples. Excessive use of “successfully synthesis” .

Row 457: “core-shell-shell composite formation was propose”  Typing error .

The authors state at line 190-191: “As a result, it was expected that the excitation and emission efficiency of core-shell-shell 190 composites were significantly enhanced.” How is the excitation efficiency enhanced? Please explain.

Luminescence

Line 359 The excitation spectra monitored with 543 nm- please replace “with” with “at”.

Eq 1 and 2 should be moved to SI or Experimental Section.

For comparison purpose and to disregard the fitting “volatility”, all decays in Figure 10a-d should be also represented in a single figure. The Table S1 should be included in the main text.

Author Response

Point 1: Major comment

At present, the field of the organic/inorganic hybrids with silica core and lanthanide organic complexes is quite inflated. The authors should correctly place their findings in the literature context, highlighting the novelty and applications. As the luminescence is apparently the proposed functionality of these compounds, the authors should compare it with the performance of similar compounds reported so far. To this aim, estimation of the quantum yield is mandatory. Otherwise, the term brightness is not supported by any of presented data. In fact, as seen from the Table S1, the variation in the average lifetimes is not significant and can be assigned mainly to fitting procedure. This make the whole synthesis useless for any possible applications compared to what is published already.

Response 1: Thanks for the reviewer’s suggestion very much. The novelty and applications have be highlighted in our manuscript. According to the reported literature, there are sufficient researches on core-shell composite materials with SiO2 as the core and inorganic materials as cladding layer, which, however, the inorganic materials usually exhibited weak luminescence property and restricted the practical application of them. Therefore, we employed the rare earth organic complexes as a coating layer, which can make full use of the "antenna effect" of organic ligand to achieve higher luminous efficiency. When the rare earth organic complex was fixed on the surface of silica sphere, the non-radiative pathways of high vibration energy loss in the ligand molecules could be largely reduced. In previous work, we have systematically studied the change of luminescence properties of the core-shell composites. Besides, the cheap silica was used as supporting cores and the nanometer magnitude terbium organic complex was employed as coating layers, which could significantly save rare earth resources and reduce the production cost. These core-shell composites can exhibit excellent luminescence in solid state, however, they usually presented poor luminescent stability properties under aqueous medium. Thus, it is necessary to improve luminescent stability. The silica-modified sol-gel technique is an ideal option to improve the solubility and luminous stability of rare earth core-shell composites. Such silica coating core-shell composites simultaneously exhibited multi-functionality.

These core-shell-shell composite materials with surface-attached abundant Si-OH molecules are easily available for conjugation with biomolecules. Therefore, they could be used for the realistic testing of the biomedical sciences.

(Page 2, Line 57-77)

(Page 2, Line 88-91)

Thanks for the reviewer’s excellent suggestion. We have compared the photoluminescence properties with the similar compounds reported so far. The photoluminescence properties of core-shell and core-shell-shell composites were compared to those of inorganic luminescent material. It is interesting to note that the emission intensities in core-shell and core-shell-shell composites are relatively stronger under the same measurement conditions. Furthermore, the core-shell-shell composites can exhibit good luminescence properties, and stability both in solid state and in aqueous solution medium.

We are very grateful for the reviewer’s suggestion. The absolute quantum yield of core-shell and core-shell-shell composites was measured. The corresponding text was modified in the revised manuscript.

(Page 12, Line 389-402)

To further investigate the photoluminescence properties of Tb(III) core-shell and core-shell-shell composites, the absolute quantum yield was determined. The absolute quantum yields of SiO2@PMDA-Si-Tb and SiO2@PMDA-Si-Tb-phen were 33.18% and 36.57%, respectively. In contrast, the corresponding absolute quantum yields of SiO2@PMDA-Si-Tb@SiO2 and SiO2@PMDA-Si-Tb[email protected]2 were 41.75% and 47.34%, respectively. When encapsulated with an amorphous silica shell, the core-shell-shell composite materials have a marked increase in absolute quantum yield. These results may be attributed to the fact that non-radiative centers existing on the surface of core-shell composites were eliminated by the shielding effect of the silica shell. It may also result from the suppression of the OH groups quenching effect by the silica shell. Meanwhile, the emission intensities of core-shell-shell composites were significantly enhanced after the coating of an amorphous silica shell. The introduction of the second ligand phen can also synergistically sensitize the luminescence of terbium ions by "antenna effect", and then largely improve the luminescent properties of the core-shell composites.

Point 2: Other comments: Synthesis and characterization

Page 3 row 126: “For the preparation of SiO2@PMDA-Si spheres, briefly, 0.20 g SiO2 spheres were ultrasonically dispersed into 20 mL ethanol forming a homogeneous dispersion, followed by dropwise addition of benzene (30 mL) containing PMDA-Si (0.26g) under continuous stirring at room temperature” Authors may need to provide additional details about the ratio between SiO2 spheres and PMDA-Si. Details such as the molar ratio alongside the weight ratio and the use or not of excess of a component may help the readability of the text.

Response 2: We are very grateful for the reviewer’s excellent suggestion. The preparation of SiO2@PMDA-Si sphere has been reasonably revised.

(Page 3, Line 127-131)

2.4. Synthesis of SiO2@PMDA-Si

The preparation of SiO2@PMDA-Si spheres with mass ratio of SiO2 to PMDA-Si was 10:13. Briefly, 0.20 g SiO2 spheres were ultrasonically dispersed into 20 mL ethanol forming a homogeneous dispersion, followed by dropwise addition of benzene (30 mL) containing PMDA-Si (0.26 g) under continuous stirring at room temperature. 

Point 3: Page 3 row 136: “The synthetic procedure for SiO2@PMDA-Si-Tb-phen was as follows, 0.15 g SiO2@PMDA-Si spheres and 0.14 g second ligand phen were dispersed in 15 mL anhydrous ethanol under continuous stirring. Then 0.20 g Tb(ClO4)3·nH2O dissolved in 5.0 mL anhydrous ethanol was added, and the mixture was heated at 50 °C for 5 h” Please confirm that you have added phen over the SiO2@PMDA-Si and afterwards Tb on top of that mixture.

Response 3: Thanks for the helpful comments and suggestions. We have revised this part according to the Reviewer’s suggestion.

(Page 4, Line 139-143)

The synthetic procedure for SiO2@PMDA-Si-Tb-phen was as follows, 0.15 g SiO2@PMDA-Si spheres and 0.14 g second ligand phen were dispersed in 15 mL anhydrous ethanol through ultrasonication. Then 0.20 g Tb(ClO4)3·nH2O dissolved in 5.0 mL anhydrous ethanol was slowly added into the above solution under vigorous stirring. The reaction mixture was heated at 50 °C for 5 h.

Point 4: Page 4 row 169: The PMDA-Si acted as a “bi-functional bridge molecular” Word inversion of the quoted term as “bi-functional molecular bridge”.

Response 4: Thanks for the helpful suggestion. We have replaced the statements of “bi-functional bridge molecular”by “bi-functional molecular bridge”.

(Page 5, Line 184)

The PMDA-Si acted as a "bi-functional molecular bridge". 

Point 5: Row 170: “connecting (…) by a hydrolysis forming Si-O-Si covalent bonds”. This sentence is ambiguous and authors should consider to revise this part.

Response 5: We are very grateful for the reviewer’s crucial comment. This sentence has been revised for more clearly description as follows:

(Page 5, Line 184-186)

The alkoxyl groups can be covalently bonded with the active hydroxyl groups on the surface of silica spheres forming Si-O-Si bonds. 

Point 6: Row 175: “In this step, because silica as supporting cores was cheap and terbium organic complex as coating layer was nanometer magnitude, the core-shell composites can significantly save rare earth resources and greatly reduce production cost” Please, detail what is the discussed step. Also, the introduction of the cost benefits in the Formation Mechanism and Design Idea is diverging from the aim of this paragraph. Please, reassign this sentence to other chapter or to place it in a different section.

Response 6: We are very grateful for the reviewer’s suggestion. It has been revised in the results and discussion. This sentence has been moved to the conclusions. 

(Page 15, Line 485-488)

In the core-shell-shell composite materials, the cheap silica was used as supporting cores and the nanometer magnitude terbium organic complex was employed as coating layers, which could significantly save rare earth resources and reduce the production cost. 

Point 7: Row 178: “Furthermore, as one end of the terbium organic complexes were fixed on the surface of SiO2 core, the vibration of the ligand molecules (PMDA-Si, phen) could be largely reduced” A clarification is needed in this sentence due to the appearance for the first time of “terbium organic complexes”. Row 179: “The core-shell composites exhibited excellent luminescence properties”. This sentence is misplaced and should be relocated. Row 181: “The other end was still in contact with the external environment, which had a quenching effect on the rare earth organic complex shell” Confuse statement, please revise it.

Response 7: Thanks for the reviewer’s comment. According to the suggestions, we have revised this part. 

(Page 5, Line 186-192)

In this process, organosilane PMDA-Si was successfully grafted onto the surface of silica sphere. Thus, the carboxyl group of PMDA-Si could coordinate to terbium ions. Additionally, the introduction of the second ligand phen could also synergistically coordinate to terbium ions by double nitrogen atoms and form terbium organic complex in the shell layer. Here, the core-shell composites were synthesized with SiO2 as the core and terbium organic complex as the shell. To enhance the solubility and the luminescent stability of the core-shell composite materials, the silica-modified core-shell-shell composites were synthesized. 

Point 8: Page 5 row 187: “inorganic substrate silica” Word inversion.

Response 8: We are very grateful for the reviewer’s excellent suggestion. We have revised the statements of “inorganic substrate silica” by “inorganic silica substrate”.

(Page 5, Line 195-196)

As an intermediate layer, the terbium organic complexes were fixed on the inorganic silica substrate. 

Point 9: Row 190: “excitation efficiency enhancement” What do the author mean by this statement ?

Response 9: We are grateful to the reviewer for pointing out the error. We have revised this part.

(Page 5, Line 197-198)

As a result, it was expected that the emission efficiency of core-shell-shell composites was significantly enhanced.

Point 10: The description of TEM data and core-shell structure should be strongly revised. Please explain use of terms such as “dark centers” and “pale edges” for describing the core-shell structures and should provide a scientific explanation regarding those dark and pale regions.

Response 10: We are very grateful for the reviewer’s carefully reading and great suggestion. The description of TEM data and core-shell structure have been revised.

(Page 5-6, Line 215-219)

After coated with terbium organic complex layer, the surface of SiO2@PMDA-Si-Tb and SiO2@PMDA-Si-Tb-phen core-shell composites became a little larger and much rougher. Figure 3b, c and e, f show the TEM images at the SiO2@PMDA-Si-Tb and SiO2@PMDA-Si-Tb-phen core-shell composites, respectively. These core-shell composites were still spherical and nonaggregated, but a slight increase in diameter than the SiO2@PMDA-Si particles. 

Point 11: Row 222: The statement that the terbium complexes were successfully synthetized is expressed twice in the same paragraph.

Response 11: Thanks for the suggestion. We have revised this part according to the reviewer’s suggestion.

(Page 6, Line 225-227)

These results seem to indicate that the SiO2@PMDA-Si-Tb and SiO2@PMDA-Si-Tb-phen core-shell composites were successfully synthesized.

Point 12: Row 230: “In order to widen the application field of core-shell composite materials, the silica-modified core-shell-shell composites were also synthesized” What applications have the authors in mind ?

Response 12: We are very grateful for the reviewer’s careful reading. For biological applications, the prepared core-shell composites should be luminescent stability and water-solubility. The silica shell can not only prevent rare earth core-shell composites from quenching of the external environment, but also improve the solubility and luminescent stability of the core-shell composite material. The core-shell-shell composites exhibited bright luminescence, high stability and good solubility, which may present potential applications in the field of bio-medical.

Point 13: Row 236: “Wherein, it showed the black core-shell composites were uniformly encapsulated into a thin gray silica shell” Please, clarify this paragraph.

Response 13: We are very grateful for the reviewer’s crucial comment. We have revised this part.

(Page 7, Line 241-244)

Wherein, it showed that the core-shell composites were uniformly encapsulated into a gray shell, and the diameter increased to approximately 210 nm corresponding to a 10 nm thick layer on the surface assigned to amorphous silica.

Point 14: Row 260: “We characterized the XPS spectra” Use interpret or analyse instead of characterization. Further, XPS provides information about the surface chemistry and not whole sample chemistry and in consequence the 3.3 Chemical analysis title is not appropriate.

Response 14: Thanks for the suggestion. We have revised this part according to the reviewer’s suggestion.

(Page 8, Line 263-266)

3.3. XPS Analysis

To further explain the incorporation of Tb-(PMDA-Si)-ClO4 and Tb-(PMDA-Si)-phen-ClO4 complexes onto the surface of SiO2, the XPS spectra of PMDA-Si, SiO2@PMDA-Si-Tb and SiO2@PMDA-Si-Tb-phen were obtained.

Point 15: Section 3.4.1: The successful synthesis of those particles is stated three times in this section and can be a conclusion of this part. The authors should rephrase the section.

Response 15: We are very grateful for the reviewer’s crucial comment. We have revised this part according to the reviewer’s suggestion.

(Page 9, Line 294-317)

3.5.1. The FT-IR Spectra of SiO2@PMDA-Si-Tb and SiO2@[email protected]2

Further to verify the purity and internal structure of core-shell SiO2@PMDA-Si-Tb and core-shell-shell SiO2@[email protected]2. The FT-IR spectra of PMDA-Si, SiO2, SiO2@PMDA-Si, SiO2@PMDA-Si-Tb, and SiO2@[email protected]2 are displayed in Figure S2. The appearance of the characteristic bands of PMDA-Si (Figure S2a) at 1633 cm−1 and 1550 cm−1 was attributed to the stretching vibration of -CONH- bond, and the stretching vibration of -C=O- (COOH) appeared at 1727 cm−1. These characteristic bands demonstrated that functional molecular bridge PMDA-Si had been successfully synthesized through the amidation reaction with pyromellitic dianhydride and 3-aminopropyltriethoxysilane. The band of SiO2 (Figure S2b) at 1098 cm−1 was attributed to the stretching vibration of Si-O-Si group, and Si–OH group was identified at 952 cm-1. In the spectra of SiO2@PMDA-Si (Figure S2c), the appearance of the characteristic band at 1718 cm-1 was ascribed as the stretching vibration of -C=O- (COOH). The characteristic bands at 1641 cm-1 and 1556 cm-1 were originated from the stretching vibration of -CONH-, which further confirmed the "functional molecular bridge" PMDA-Si onto the surface of SiO2. The FT-IR spectrum of SiO2@PMDA-Si-Tb core-shell composites (Figure S2d) shows the characteristic bands of -C=O- (COOH) bonds at 1714 cm-1. There was an obvious change in the absorption band compared with SiO2@PMDA-Si, indicating that Tb3+ ions coordinated to the oxygen atoms of carbonyl group. In addition, observed characteristic infrared band at 1147, 925 and 622 cm-1 was assigned to the stretching and bending vibration modes of the perchlorate (ClO4-) group. As can be observed in the FT-IR spectrum of the SiO2@[email protected]2 core-shell-shell composites (Figure S2e), there was a significant change between the spectra of SiO2@PMDA-Si-Tb and SiO2@[email protected]2. After coating the SiO2 shell, the intensity of the characteristic -CONH- and -C=O-(COOH) absorption bands was obviously decreased. The surface Si-OH groups play an important role for the functionalization and their affinity with bio-molecules, which is usable in the bio-medical field.

Point 16: Section 3.5 : The authors note the 2Theta values of the diffraction peaks but do not mention the observed shift towards higher angles between the two samples. Excessive use of “successfully synthesis” .

Response 16: We are very grateful for the reviewer’s crucial comment. We have revised this part according to the reviewer’s suggestion.

(Page 10, Line 333-358)

3.6. X-ray Diffraction Analysis

The composition and structure of the powder samples were examined by XRD. Figure 8(A) shows the XRD patterns of SiO2 core, SiO2@PMDA-Si, SiO2@PMDA-Si-Tb and SiO2@[email protected]2, respectively. For SiO2 directly formed from Stöber method (Figure 8(A)a), there were two broad bands centered at 2θ = 7-8° and 2θ = 23°, which were identical with the standard XRD pattern for amorphous SiO2. After the bi-functional organosilane PMDA-Si was grafted onto the surface of SiO2, no other phase was detected from the comparison of SiO2 in Figure 8(A)b. For SiO2@PMDA-Si-Tb core-shell composite in Figure 8(A)c, the diffraction pattern also consisted of two broad band appeared at 2θ = 7-8° and 2θ = 23°, and the intensity of the diffraction band became weaker due to the coating of Tb-(PMDA-Si)-ClO4 complexes. Besides, there were several sharp peaks at about 7.3°, 8.3°, 13.8°, 20.7°, 22.1°, and 23.8°, indicating that Tb-(PMDA-Si)-ClO4 complexes coated on the surface of SiO2 core. As illustrated in Figure 8(A)d, the two broad bands of SiO2@[email protected]2 core-shell-shell composites appeared at 2θ = 7-8° and 23° were in coincidence with the stander XRD pattern for SiO2 core (Figure 8(A)a), which further illustrated that an amorphous SiO2 was coated onto the surface of SiO2@PMDA-Si-Tb. 

After the incorporation of the phen and SiO2 shell, the growth of Tb-(PMDA-Si)-phen-ClO4 and SiO2 shell onto the core-shell and core-shell-shell composites has been characterized by XRD. Figure 8(B) shows the XRD patterns of SiO2 core, SiO2@PMDA-Si, SiO2@PMDA-Si-Tb-phen and SiO2@[email protected]2. Compared with the diffraction pattern of SiO2 core (Figure 8(B)a), several small sharp diffraction peaks at about 7.9°, 8.9°, 12.4°, 18.7°, 21.1°, 23.8° and 27° were observed for the SiO2@PMDA-Si-Tb-phen composites (Figure 8(B)c). It means that the existence of Tb-(PMDA-Si)-phen-ClO4 on the surface of SiO2 core. The diffraction pattern of the SiO2@[email protected]2 composites (Figure 8(B)d) displayed the disappearance of several small diffraction peaks. Moreover, the broad bands centered at 2θ = 23° shifted to a little larger diffraction angles compared with the pure SiO2 core. This phenomenon may be attributed to the growth environment of the silica core and the silica shell was different.

Point 17: Row 457: “core-shell-shell composite formation was propose” Typing error .

Response 17: We are grateful to the reviewer for pointing out the error. We have revised this part.

(Page 14, Line 476-477)

The formation mechanism of core-shell and core-shell-shell composite was proposed.

Point 18: The authors state at line 190-191: “As a result, it was expected that the excitation and emission efficiency of core-shell-shell composites were significantly enhanced.” How is the excitation efficiency enhanced? Please explain.

Response 18: We are grateful to the reviewer for pointing out the error. We have revised this part.

(Page 5, Line 197-198)

As a result, it was expected that the emission efficiency of core-shell-shell composites was significantly enhanced.

Point 19: Luminescence

Line 359 The excitation spectra monitored with 543 nm please replace “with” with “at”.

Response 19: Thanks for the suggestion. We have revised this part.

(Page 11, Line 374)

The excitation spectra monitored at 543 nm were investigated in Figure 9a and c.

Point 20: Eq 1 and 2 should be moved to SI or Experimental Section.

Response 20: We are very grateful for the reviewer’s excellent suggestion. We have moved this part to Experimental Section.

(Page 4, Line 172-178)

2.8. Lifetime Analysis

The photoluminescence lifetime analysis for the representative emission of Tb3+ ions was performed using an Edinburgh FLS980 spectrophotomete at room temperature. The decay curve can be fitted into a biexponential function (1). The lifetime values of the excited state terbium ion (5D4) could be calculated using the following equation (2), where τ1 and τ2 stand for the slow and fast terms of the luminescent lifetime, respectively, and A1 and A2 are the corresponding pre-exponential factors.

I(t) = I0 + A1 exp(-t11) + A2 exp(-t22)

(1)

(τ) = (A1τ12 + A2τ22)/(A1τ1 + A2τ2)

(2)

Point 21: For comparison purpose and to disregard the fitting “volatility”, all decays in Figure 10a-d should be also represented in a single figure. The Table S1 should be included in the main text.

Response 21: We are very grateful for the reviewer’s excellent suggestion. We have revised this part.

(Page 13-14, Line 428-445)

3.8. The Photoluminescence Lifetime 

Figure 11. Decay curve of SiO2@PMDA-Si-Tb (a), SiO2@PMDA-Si-Tb@SiO2 (b), SiO2@PMDA-Si-Tb-phen (c) and SiO2@[email protected]2 (d).

In order to study the details of photoluminescence property for terbium core-shell and core-shell-shell composites, the photoluminescence decay curve of SiO2@PMDA-Si-Tb, SiO2@PMDA-Si-Tb-phen and SiO2@PMDA-Si-Tb@SiO2, SiO2@PMDA-Si-Tb-phen@SiO2 composites was recorded in Figure 11. The decay curve can be well fitted into a biexponential function. Herein, calculated average lifetime of two terbium core-shell composites SiO2@PMDA-Si-Tb and SiO2@PMDA-Si-Tb-phen were 927.32 μs and 960.85 μs, respectively. Moreover, the average lifetime of two terbium core-shell-shell composites SiO2@PMDA-Si-Tb@SiO2 and SiO2@PMDA-Si-Tb-phen@SiO2 were 1086.28 μs and 970.18 μs, respectively. The increase of the photoluminescence lifetime for two terbium core-shell-shell composites showed that the quenching from outside particles was strongly reduced after the growth of a silica shell around the core-shell composites. The detail values are listed in Table 2. Two terbium core-shell-shell composites both have relatively long lifetimes under UV excitation.

Table 2. The lifetime parameters obtained from the biexponential fitting to the photoluminescence decay values of Tb(III) core-shell and core-shell-shell composites.

Composites

A1

A2

τ1 (µs)

τ2 (µs)

τ (µs)

R2

SiO2@PMDA-Si-Tb

620.558

2817.895

409.1

975.2

927.32

0.999

SiO2@PMDA-Si-Tb-phen

2844.215

1319.362

1009.5

126.9

960.85

0.999

SiO2@[email protected]2

598.777

1044.710

1373.6

805.4

1086.28

0.9999

SiO2@[email protected]2

442.512

1256.689

424.8

1048.2

970.18

0.998

Reviewer 3 Report

The preparations carried out in a methodical manner are described and the results are presented clearly but not contrasted with what is described in the literature. In all the sections there is no discussion of the results. The authors should complement all this.

Author Response

Point 1: 
The preparations carried out in a methodical manner are described and the results are presented clearly but not contrasted with what is described in the literature. In all the sections there is no discussion of the results. The authors should complement all this.

Response 1: We are very grateful for the reviewer’s suggestion. We have complemented this part in all the sections. At present, there are sufficient researches on core-shell luminescence materials with SiO2 as the core and inorganic materials as cladding layer. However, the inorganic materials usually exhibited weak luminescence property and restricted their practical application. Therefore, in order to obtain excellent luminescence functional materials, we employed the rare earth organic complexes as the coating layer, which can make full use of the "antenna effect" of organic ligand to achieve higher luminous efficiency. In particular, luminescence applications of rare earth organic core-shell materials are very attractive because of their superior luminescence intensity. Among the number of luminescence materials, there are two main methods have been used in the preparation of core-shell materials. One is direct deposition method, and the other is silane coupling agent method. But the core-shell materials were not stable using the former method. It was easy to collapse under the effect of ultrasound. Therefore, silane coupling agent method is considered as an effective and popular strategy to prepare the core-shell luminescence materials. Here, the silane coupling agent acted as a "bi-functional molecular bridge" connects SiO2 spheres and rare earth complexes together by covalent bond. The core-shell composite materials are stable and the thickness of the shell is easily controlled. As a result, these kinds of core-shell materials have high luminescence intensity and fluorescence quantum efficiency in comparison with conventional phosphor materials. In order to further enhance the luminescent stability and the solubility of the core-shell composite materials in aqueous solution, we proposed a cost-effective synthesis procedure for the preparation of the core-shell-shell luminescence materials. The photoluminescence properties of core-shell-shell composites were compared to those of core-shell luminescent material. It is interesting to note that the emission intensities in core-shell-shell composites were relatively stronger. Meanwhile, the core-shell-shell composites exhibited better luminescence stability in the aqueous medium. Especially, the core-shell-shell structure was not collapsed even after three days. Therefore, these core-shell-shell composite materials exhibit good luminescence properties and stability in aqueous solution medium, which could be used for the realistic testing of the biomedical sciences.

Round  2

Reviewer 1 Report

The manuscript can be accepted accepted for publication in its current form. 

Reviewer 2 Report

Please, correct the typos, adjust the flow of the paper and improve the English language and grammar.

Reviewer 3 Report

The new version has been improved a lot.